# Preparation Scheme Optimization of Thermosetting Polyurethane Modified Asphalt

**DOI:** 10.3390/polym15102327

**Published:** 2023-05-16

**Authors:** Min Sun, Shuo Jing, Haibo Wu, Jun Zhong, Yongfu Yang, Ye Zhu, Qingpeng Xu

**Affiliations:** 1School of Transportation Engineering, Shandong Jianzhu University, Jinan 250101, China; jingshuo99@163.com; 2MCC Road & Bridge Construction Co., Ltd., Jinan 250031, China; 13561669518@163.com; 3Shandong Transportation Institute, Jinan 250031, China; zhongjun@sdjtky.cn; 4Shandong Provincial Communications Planning and Design Institute Co., Ltd., Jinan 250031, China; zhuye1223@163.com; 5Jinan Urban Construction Group Co., Ltd., Jinan 250031, China; xuqingpeng2005@163.com

**Keywords:** road engineering, thermosetting polyurethane asphalt, orthogonal experiment, preparation plan

## Abstract

To solve the issue of the poor temperature stability of conventional modified asphalt, polyurethane (PU) was used as a modifier with its corresponding curing agent (CA) to prepare thermosetting PU asphalt. First, the modifying effects of the different types of PU modifiers were evaluated, and the optimal PU modifier was then selected. Second, a three-factor and three-level L9 (3^3^) orthogonal experiment table was designed based on the preparation technology, PU dosage, and CA dosage to prepare the thermosetting PU asphalt and asphalt mixture. Further, the effect of PU dosage, CA dosage, and preparation technology on the 3d, 5d, and 7d splitting tensile strength, freeze-thaw splitting strength, and tensile strength ratio (TSR) of the PU asphalt mixture was analyzed, and a PU-modified asphalt preparation plan was recommended. Finally, a tension test was conducted on the PU-modified asphalt and a split tensile test was performed on the PU asphalt mixture to analyze their mechanical properties. The results show that the content of PU has a significant effect on the splitting tensile strength of PU asphalt mixtures. When the content of the PU modifier is 56.64% and the content of CA is 3.58%, the performance of the PU-modified asphalt and mixture is better when prepared by the prefabricated method. The PU-modified asphalt and mixture have high strength and plastic deformation ability. The modified asphalt mixture has excellent tensile performance, low-temperature performance, and water stability, which meets the requirements of epoxy asphalt and the mixture standards.

## 1. Introduction

As the main form of road surface in China, asphalt pavement has distinct advantages, such as a good driving comfort, excellent road performance, strong environmental adaptability, and significant economic benefits [1,2]. As a binder for asphalt pavement, asphalt has an excellent bonding performance. However, with the gradual improvements in road construction at all levels in China, the traffic volume and vehicle axle load have sharply increased, along with road disease also gradually increasing as a result, meaning that the mechanical properties of asphalt mixture can no longer meet the developmental needs for high-quality roads. Therefore, modified asphalt technologies have been proposed [3,4,5]. At present, modified asphalt technologies, such as SBS modification, rubber powder modification, graphene modification, and composite modification have been rapidly developed. Modified asphalt technology can improve the water damage resistance, as well as the high and low temperature performances of asphalt [6,7,8]; however, there are still issues regarding temperature stability and durability [9,10]. Epoxy asphalt has the characteristics of a thermosetting material, with excellent strength, stiffness, durability, and temperature stability, but there are limitations in terms of its high price, harsh construction conditions, and insufficient toughness [11,12,13]. Therefore, it is imperative to develop novel modified asphalt materials.

PU is a green modified asphalt material, and is a multi-block polymer that can adjust the type and proportion of isocyanate, polyol, and other additives to design different structures. With its high degree of designability, it can be utilized used across various industrial production fields, such as plastics, films, adhesives, synthetic leather, and elastomers [14,15,16]. Currently, the primary application of PU in the domain of road pavement mainly focuses on two research directions, with one of them involving using PU to replace asphalt as a binder. The prepared PU mixture has high strength, high-temperature performance, low-temperature performance, excellent water stability, and fatigue resistance. It can substantially decrease the layer thickness of the pavement structure, prolong the service life of the pavement, save energy, reduce emissions, and is environmentally friendly [17,18,19]. The second involves using PU as the modifier for the preparation of PU asphalt. Due to the flexible formulation, diverse product forms, and excellent performance of the PU materials, the types and characteristics of PU-modified asphalt differ. The compatibility, storage stability, and economy of PU as an asphalt modifier must therefore be considered [20,21,22].

To explore the compatibility between polyurethane and asphalt and its influence on asphalt performance, in recent years, researchers have explored the modification mechanisms and compatibility of PU asphalt through molecular simulation, infrared spectroscopy (FI-TR), and fluorescence microscopy. A study on molecular dynamics has shown that the PU modifier has the best effect on asphalt modification at 140 °C [23]. The PU modifier exhibits a positive direct effect on the swelling and dispersion between aromatics–aromatics and saturated–saturated components in asphalt, decreasing the congregation of the two molecules. At the same time, the PU content and shear temperature will also affect the compatibility of asphalt and PU. The characteristic peak of asphalt aging in FT-IR spectra was found to have weakened. It was considered that the isocyanate group reacted with active hydrogen in asphalt, leading to cross-linking, and the PU chain was subsequently broken down and degraded after aging, which further enhanced the compatibility of PU and asphalt [24]. The microstructure of PU composite modified asphalt can be investigated with scanning electron microscopy and FT-IR [25], which are both helpful to understand the modification mechanism and corresponding properties of this asphalt. The results of dynamic thermodynamic analysis and differential scanning calorimetry showed that the addition reaction between the polyisocyanate in the polyurethane prepolymer and the aromatic compounds in base asphalt improved the performance of asphalt [20].

At present, PU asphalt is mainly composed of thermoplastic PU (TPU) asphalt and composite modified polyurethane asphalt. In the field of thermoplastic PU (TPU) asphalt, Zhang et al. [26] studied a new type of PU thermoplastic (PUTE) as an asphalt modifier and conducted a series of experimental studies to explore the effect of PU content on asphalt performance. The results indicated that PUTE can significantly enhance the high-temperature, low-temperature, and mechanical performance of asphalt, and the optimum content was found to be 11%. Salas et al. [27] used PU foam to prepare PU-modified asphalt and found that, in comparison to conventional polymer-modified asphalt mixtures, PU-modified asphalt has improved stability and low-temperature deformability. Jin et al. [28] used TPU as a modifier to be incorporated into base asphalt and found that TPU-modified asphalt had better properties, such as ductility, softening point, needle penetration, and rotational viscosity after the experimental analysis of the chemical, microstructural, and rheological properties of the modified asphalt, and pointed out that polyester-based TPU modifiers have better high-temperature properties than polyether-based TPU modifiers.

In the field of composite modified polyurethane asphalt, Zhang et al. [29] prepared an epoxy/PU (EPU) modified asphalt using PU and epoxy resin. The mechanical properties and microstructure of EPU-modified asphalt were evaluated by FT-IR testing, tensile tests, and scanning electron microscopy. It was found that EPU has good mechanical strength, flexibility, and storage stability. Yan et al. [30] prepared bone glue/polyurethane composite modified asphalt (CMA). The relationship between the modifier content and CMA’s conventional properties and rheological properties was determined using the response surface method. It was found that the low-temperature crack resistance, water stability, and dynamic stability of CMA mixtures were enhanced. Jin et al. [21] analyzed the rheological properties, chemical properties, and microstructure of PU asphalt by adding a mixture of PU and RA rock asphalt. The results showed that PU composite modified asphalt had a good performance in terms of permeability, ductility, softening point, and rotational viscosity. Moreover, the isocyanate reacted with phenol and carboxylic acid in modified asphalt to further improve the performance.

Most current research on PU asphalt is either on a certain modifier or composite modification, and there are a lack of comparisons among various modifiers. There are still issues that remain from its utilization, such as high construction temperatures and poor temperature stability [31]. Therefore, thermosetting polyurethane modified asphalt has been studied by several scholars. Zhang et al. [32] determined the appropriate formula for a thermosetting PU (TS-PU) modified binder through a series of laboratory tests, studied the comprehensive properties of asphalt modifiers containing various TS-PU contents, and determined the optimum content of modifiers. It has also been shown that a TS-PU modified modifier is superior to epoxy modifiers in terms of the high-temperature rutting resistance and tensile strength, flexibility, and cost. Jia [33] evaluated the performance of a thermosetting PU (TS-PU) asphalt mixture and a TPU asphalt mixture through conducting a series of tests. The results showed that the two kinds of PU asphalt mixture have better flexibility and low-temperature crack resistance than epoxy asphalt mixtures. The performance difference between these PU asphalt mixtures is also significant, indicating that the PU asphalt modifier is of great significance for obtaining PU asphalt pavement with customized performance. Yang et al. [22] also found that thermosetting PU can enhance the low-temperature flexibility through BBR tests. In addition, although the tensile strength of TSPU is lower than that of epoxy asphalt, when the TSPU content exceeds 40%, the fracture energy is comparable to that of epoxy asphalt through direct tensile tests. In summary, thermosetting PU-modified asphalt can improve the temperature stability, flexibility, and mechanical properties of asphalt, and has a greater development value.

Therefore, this study conducted a comparative analysis of the tests of each component material of thermosetting PU asphalt and optimized the best PU modifier. Based on the three-factor and three-level orthogonal test of the preparation process, PU content, and CA content, a PU asphalt mixture was prepared. Then, the 3d, 5d, and 7d splitting tensile strength, freeze-thaw splitting strength, and TSR of the PU asphalt mixture were all tested. The effects of various factor levels and combinations on the basic mechanical properties of PU asphalt mixture were analyzed, and the best preparation scheme of PU-modified asphalt was recommended, which provides a reference for future research on thermosetting PU asphalt.

## 2. Materials and Methods

### 2.1. Materials

The base asphalt was selected as the raw material for PU asphalt, and four kinds of PU and two kinds of CA were used to modify the original asphalt by single mixing. The technical indexes of the base asphalt are shown in Table 1.

The aggregate was provided by the Shandong Provincial Transportation Planning and Design Institute, composed of limestone gravel and manufactured sand, which was mainly divided into 0–4 mm, 4–6 mm, 6–12 mm, 12–16 mm, and mineral powder. The aggregate indexes of each grade are displayed in Table 2. The AC-13 PU asphalt mixture gradation composition is shown in Figure 1.

Four kinds of PU modifiers produced by the Wanhua Group were selected to prepare PU-modified asphalt with CA. These modifiers were labelled as GXJ-1, GXJ-2, GXJ-3, and GXJ-4. The material composition characteristics of the modifier and its corresponding CA are shown in Table 3.

The volumetric properties of mixtures are shown in Table 4.

### 2.2. Test Method

#### 2.2.1. Splitting Strength and Height Test of PU Specimens

At room temperature, the PU was mixed with the CA, and the standard Marshall test specimen—with a diameter of 101.6 mm and a height of 63.5 mm—was made as according to the design grading requirements [34]. After the mixture was mixed, Marshall compaction was conducted. The number of compactions made was fifty times for each side, and the compaction temperature was 120 °C. After the specimens were cured for 3d, 5d, and 7d at room temperature, the splitting tensile test was conducted with reference to the test regulations [34]. The 3d, 5d, and 7d splitting tensile strengths of the different polyurethane modifier materials were assessed as performance indicators for evaluating the polyurethane materials.

The PU in the mixture reacts with the water molecules in the air to form CO_2_ gas, which cannot be discharged due to the low porosity, causing the volumetric expansion of the specimen. After 7 days of curing, the height of the PU specimen was measured with a vernier calliper. In comparison to the height of the initial specimen, Figure 2 shows the measurement method for the PU specimen height. The change in height can reflect the expansion phenomenon of the specimen. When H′ = H, there is no bulge in the specimen. When H′ > H, the bulge of the specimen will increase its height.

#### 2.2.2. Optimization of PU

Four kinds of PU modifiers were modified through single blending to prepare the different modified asphalts. First, base asphalt and PU (GXJ-1, GXJ-2, GXJ-3, and GXJ-4) were preheated to 120 °C. Second, PU (mass fractions of 10%, 30%, and 50%) was added to the base asphalt, placed in a heating plate, and heated to 120 °C, and then sheared for 10 min with high-speed shear at 4000 rpm. After shearing was completed, CA-1 and CA-2 (mass fractions of 12% and 6%, respectively) were slowly added to continue shearing for 10 min to prepare the PU asphalt. After the preparation was completed, it was mixed with an aggregate to form Marshall specimens. The forming process is shown in Figure 3. The 3d, 5d, and 7d splitting tensile strengths of the different PU-modified asphalts were assessed; the splitting test is shown in Figure 4. A PU modifier was then selected.

#### 2.2.3. Orthogonal Experiment of PU-Modified Asphalt

After optimizing a PU modifier, an L9 (3^3^) orthogonal test table based on the preparation method, PU content, and CA content was designed to prepare the PU asphalt and mixture. The influences of PU content, CA content, and the preparation method on the 3d, 5d, and 7d splitting tensile strength, freeze-thaw splitting strength, and TSR of PU asphalt mixture were analyzed. The orthogonal factors and horizontal are listed in Table 5, the L9 (3^3^) orthogonal test table is listed in Table 6, and the preparation process of PU asphalt is laid out in Table 7.

To further optimize the experimental conditions for the preparation of polyurethane asphalt, the effects of polyurethane content, curing agent content, and the preparation process on the mechanical properties of polyurethane asphalt were studied using the response surface optimization method. The three-factor three-level response surface test was designed in the Design-Expert.V8.0.6.1 software to obtain the best preparation scheme for polyurethane-modified asphalt.

#### 2.2.4. Tensile Test

In accordance with the GB/T 528-1998 standard, a tensile performance test [35] was conducted. After preparing the polyurethane asphalt, it was molded into a tensile sample in a PTFE mold and cured at room temperature for one week. The tensile test of the specimen was conducted using a WDL-2000 electronic tensile tester. The tensile strength and the elongation at break of the polyurethane asphalt were assessed using a WDL-2000 machine at a tensile rate of 500 mm/min and a temperature of 23 °C. The dimensions of the tensile specimen are shown in Figure 5. The entire experimental process was conducted as shown in Figure 6.

## 3. Results and Discussion

### 3.1. PU-Modified Asphalt Optimization

The 3d, 5d, and 7d splitting tensile strength results of the different PU modifier materials are shown in Figure 7, and the changes in the expansion height of the specimen are indicated in Figure 8.

It is clear from Figure 7 that the splitting strength of polyurethane mixtures increases accordingly with the increases in the curing time. This indicates that the longer the curing time of the polyurethane mixture, the more complete the curing reaction of the polyurethane, resulting in a higher strength of the mixture. The strength and formation rate of the four PU specimens are in the following order: PU-1 > PU-3 > PU-2 > PU-4, indicating that the PU-1 mixture has the highest strength and a rapid curing reaction. The height of expansion of the same polyurethane mixture with increasing curing time is shown in Figure 8. The expansion heights of the specimens are in the following order: PU-4 > PU-2 > PU-3 > PU-1, indicating that with the curing reaction, the polyurethane mixture continuously releases CO_2_ gas. This makes the specimen expand, and with the volumetric expansion of PU-1 being the smallest, this was deemed as the best in comparison to other polyurethane mixtures.

The PU content was determined to be 10%. Under the condition of 12% CA, the splitting tensile strengths of PU asphalt with different modifier materials at 3d, 5d, and 7d were assessed. The results of this test are displayed in Figure 9.

The data in Figure 9 indicates that the strength of the polyurethane asphalt specimens increases with the corresponding increases in the polyurethane content. The 7d splitting strength of the PU asphalt specimens is in the following order: PU-1 > PU-2 > PU-3 > PU-4. The 7d strength of PU-1, PU-2, PU-3, and PU-4 asphalt with 10% PU content was found to be 95%, 50%, 45%, and 37% higher than that of base asphalt, respectively, showing that PU can dramatically increase the strength of the asphalt specimens. PU-1 was found to have the highest strength and the greatest modification effectiveness.

The 3d, 5d, and 7d splitting tensile strengths of PU asphalt with different dosages and CA dosages were also evaluated. The test data are displayed in Figure 10.

It is clear from Figure 10 that the splitting strength of 50% PU content is 31% and 97% higher than that of 30% and 10% PU, respectively, indicating that higher contents of PU lead to higher mechanical strengths. With improvements in the CA content, the splitting strength of the PU asphalt specimens exhibited a trend of initially increasing and then decreasing, indicating that the high content of CA may lead to the early curing of PU asphalt in the forming specimen along with a loss in strength.

In summary, the quantities of CA and PU affects the properties of PU asphalt, and the preparation method needs to be further optimized. Therefore, the optimal preparation scheme of PU asphalt was determined by designing the three-factor and three-level orthogonal test of the preparation process, PU dosage, and CA dosage.

### 3.2. Range Analysis of the Orthogonal Test

The test results of the 3d, 5d, and 7d splitting strength, freeze-thaw splitting strength, and TSR of the different orthogonal combinations are shown in Table 8. The impacts of the various factors on the splitting strength of the PU asphalt mixture are shown in Figure 11, and the influence of the freeze-thaw splitting strength and the TSR of the PU asphalt mixtures are displayed in Figure 12.

It can be observed from Figure 11 that the splitting strength of polyurethane mixtures at all ages increases gradually with increases in the PU dosage. When the PU content is 60%, the splitting strength of 3d, 5d, and 7d is 1.31 times, 1.31 times, and 1.29 times higher than that of 40% PU, respectively. It can be seen that the PU content has a great influence on the splitting strength. On one hand, with increases in the PU content, the proportion of PU in the modified asphalt increases, and the mechanical properties of PU lead to an increase in the modified asphalt strength. On the other hand, a high content of PU can play a connecting role in the cross-linked network architecture developed by the PU asphalt, enhance the cohesion of the modified asphalt, and increase the splitting strength of the specimen. The 7d splitting strength of the PU asphalt specimen was found to be the highest, being 64.3% higher than that of the 3d curing condition. The analysis concluded that the strength of PU asphalt was affected by its gradual reaction with the moisture in the air. The early formation strength was low; however, with increases in the curing time, the curing reaction proceeds to gradually increase the strength.

It can be seen in Figure 12 that the freeze-thaw splitting strength increases with increases in the PU content. The freeze-thaw splitting strength at 50% and 60% PU is 34% and 41% higher than that at 40% PU, respectively, indicating that 50% and 60% PU can significantly improve the low-temperature performance and water stability of PU asphalt. With increases in the PU content, the TSR exhibited a trend of first increasing and then decreasing. When the PU content was 50%, the TSR reached its the maximum, indicating that the high content of PU will affect the water stability of the PU asphalt specimen. The best water stability and low-temperature properties of the polyurethane asphalt specimens were therefore obtained when the polyurethane content was 50%.

It can be also seen in Figure 11 how the preparation method has distinct effects on the splitting strengths at different ages. In the initial stages of the curing, there were no significant differences found between the precast method and the post-mixing method in terms of the strength. As the reaction progressed, the post-mixing method accelerated the reaction of the PU asphalt, and the strength of the specimen was found to be greater than that of the prefabricated method. After the reaction was completed, the 7d splitting strength of the specimens prepared by the prefabricated method were found to have increased by 3.2% and 31%, respectively, for the post-mixing method and the pre-reaction method. The pre-reaction method exhibited low strengths at each age; therefore, this method was excluded in subsequent testing. It can be concluded from Figure 12 that the prefabrication method has a significant impact on the freeze-thaw splitting strength and the TSR, indicating that the PU-modified asphalt prepared by the prefabrication method can provide a better water stability and low-temperature performance to the specimens.

It can be concluded from Figure 11 that the splitting strength of the specimens decreases with increases in the CA content. When the CA content changed from 3% to 6%, the strength of the specimen remained practically unchanged. When the content of CA increased to 12%, the splitting strength of the specimen decreased by 13%. Increases in the CA content led to the rapid formation of PU asphalt strength, and the specimen cannot therefore form strength in the subsequent curing process. Therefore, when the CA content was 3–6%, the reaction speed of the specimen was able to be improved without a loss in the strength. It can also be observed in Figure 12 how the freeze-thaw splitting strength and TSR of PU asphalt decreased as the CA content increased. Therefore, the water stability and low-temperature performance of PU asphalt can be enhanced when the CA content is less than 6%.

### 3.3. Orthogonal Experimental Difference Analysis

To further explain the effects of the various factors on PU asphalt, the results in Table 8 were used to perform ANOVA on the splitting strength, 7d freeze-thaw splitting strength, and TSR of the 3d, 5d, and 7d PU asphalt specimens. The results are displayed in Table 9. From the ANOVA results in Table 9, it can be concluded that the factors affecting the significance of the 5d and 7d splitting strength and freeze-thaw splitting strength are in the following order: preparation process > PU dosing > hardener dosing; the factors affecting the significance of 3d splitting strength are in the following order: PU dosing > preparation process > hardener dosing; and the factors affecting the significance of the TSR are in the following order: hardener dosing > PU dosing > preparation process.

Two factors, the PU content, and the preparation process, have significant effects on the splitting strength and freeze-thaw splitting strength, while the amount of curing agent has a significant effect on the TSR. Based on the range analysis and variance analysis methods, the better ratio was found to be A2B1C1 and A3B1C1, that is, the polyurethane contents were 50% and 60%, the prefabrication preparation method was used, and the curing agent content was 3%.

### 3.4. Response Surface Model

The experimental conditions of the three factors and the three levels were optimized using the Design-Expert software. Based on the measured data, the response surface fitting equations *Y*_1_, *Y*_2_, and *Y*_3_ of the 7d splitting strength, freeze-thaw splitting strength, and splitting tensile strength were obtained through interpolation.
(1)$$Y_{1}=2.02+0.29A-0.22B-0.15C-0.037\mathrm{AB}+0.023\mathrm{AC}+0.084\mathrm{BC}$$
(2)$$Y_{2}=1.53+0.28A-0.32B-0.15C-0.13\mathrm{AB}+0.070\mathrm{AC}+0.022\mathrm{BC}$$
(3)$$Y_{3}=80.60+1.05A-1.26B-3.57C+0.16\mathrm{AB}-1.52\mathrm{AC}-0.26\mathrm{BC}$$

Taking the response surface of the prefabricated method as an example, as shown in Figure 13, model variance analysis and reliability analysis were conducted, the results of which are shown in Table 10.

From Table 10, it can be seen that the *p*-values of the three types of models were less than 0.05, which indicates that the models fit well and can accurately reflect the effects of each factor on the 7d splitting strength, freeze-thaw splitting strength, and TSR. Furthermore, the experimental results were found to be consistent with the results of the orthogonal analysis, indicating that the response surface optimization analysis has a good reliability.

To determine the optimal ratio of PU asphalt, the maximum splitting strength and freeze-thaw splitting strength were taken as the optimization objectives. In Design-Expert, the optimal factor level obtained using the numerical tool in the optimization module is shown in Table 11.

Five groups of specimens were prepared according to the optimal scheme, and the base asphalt was used as the control group to verify its mechanical properties. First, the polyurethane was heated to 60 °C in the oven and the base asphalt was heated to 120 °C. After heating was completed, it was sheared at 120 °C and 4000 rpm for 10 min. Then, a 3.5% curing agent was added, and shearing was continued for 5 min. Finally, the PU asphalt was mixed with the aggregate to prepare the mixture, and the Marshall test specimen was formed for testing. The specific procedures conducted are exhibited in Figure 14. The property indexes of the comparison epoxy asphalt mixture are shown in Figure 15.

It can be concluded from Figure 15 that the splitting strength, freeze-thaw splitting strength, and TSR of a PU asphalt mixture are 12.9%, 21.6%, and 5.7% higher than those of an epoxy asphalt mixture, and 55.2% and 53.7% higher than those of an asphalt mixture, respectively, while the TSR does not increase. This indicates that the mechanical properties and low-temperature performance of PU asphalt are markedly better than those of epoxy asphalt and SBS-modified asphalt, while the water stability has also been improved; it has excellent mechanical properties, low-temperature performance, and water stability.

### 3.5. Tensile Properties

PU asphalt is a thermosetting material, and its mechanical properties are usually evaluated using direct tensile tests. The effects of various PU and CA contents on the mechanical performance of PU asphalt were determined through conducting tensile tests. The samples are shown in Figure 16, and the test results are presented in Figure 17. When the amount of PU was constant, an increase in the CA content made the tensile strength increase initially and then decrease, and the elongation at the break also decreased, which indicates that a certain amount of CA can improve the mechanical strength of PU asphalt but will adversely affect its flexibility. Therefore, the mass fraction of CA was set to 3%.

It can be seen from Figure 17 that when the mass fraction of CA was set to 3%, and with the increase in PU content from 40% to 50%, the tensile strength of the specimen increased from 1.13 MPa to 2.78 MPa. The increment was 1.65 MPa, while the elongation at break increased from 254% to 314%, representing an increase of 60%. With an increase in PU content from 50% to 60%, the tensile strength of the specimen further improved from 2.78 MPa to 3.37 MPa, with an increment of 0.59 MPa, and the elongation at break increased from 314% to 331%, with an increment of 17%.

It can be concluded that, with increases in the PU content, the cross-linking network inside PU asphalt becomes denser. After the complete curing reaction, a stable three-dimensional network structure was formed inside PU asphalt, which greatly improves its mechanical strength. When the PU content exceeds 50%, the tensile performance increment decreases, indicating that a PU content of about 50% is the critical point for the formation of the three-dimensional network structure inside PU asphalt.

### 3.6. Failure Surface Analysis

The PU mixture specimens were prepared as according to the optimal scheme, and the failure surfaces of the specimens were analyzed. The splitting failure surface of the specimen is shown in Figure 18a. The splitting failure surface after freeze-thaw is shown in Figure 18b.

It can be seen in Figure 18 that after the splitting test of the specimen without freeze-thaw, there were many broken aggregates on the failure surface of the specimen, indicating that failure mainly occurs in the aggregate, while the fracture form is aggregate failure, indicating that PU-modified asphalt has an excellent bond strength. The formation of a thicker binder film enhances the strength of the binder–aggregate interface and improves the mechanical strength of the mixture.

In comparison to the failure surface of the specimen before freeze-thaw, there were fewer aggregates broken on the failure surface after freeze-thaw, while the crack extended along the interface between the binder and the aggregate, indicating that in the low-temperature environment, the effect of water accelerates the failure of the binder–aggregate interface, the damage occurs mainly along the interface between the binder and the aggregate, and the fracture form is interface damage. The splitting failure simulation diagram of the specimen is shown in Figure 19.

Under the action of water erosion, water molecules entered the interface between the binder and the aggregate, which reduced the interface’s strength. When the bonding ability of the binder was low, water erosion greatly reduced the bonding strength of the interface, and the binder and the aggregate were therefore separated. Therefore, it is of particularly importance to improve the bonding ability of the binder.

As shown in Figure 19, the PU asphalt–aggregate interface strength was decreased in water and low-temperature environments, which adversely affects the interface’s strength; however, some aggregate damage still remains in the polyurethane asphalt specimens, and the splitting strength after freeze-thaw was still significantly better than that of the asphalt mixture indicating that PU asphalt has an excellent bonding performance and water stability.

## 4. Conclusions

The main conclusions of this paper are summarized as follows:(1)Under the same preparation conditions and 10% PU content, the selected PU-1, PU-2, PU-3, and PU-4 materials can improve the mechanical properties of asphalt to varying degrees. Among them, PU-1 has the most significant improvement, at 95% higher than that of base asphalt.(2)Based on the orthogonal tests and variance analysis, the influence of the related factors on the splitting tensile strength and freeze-thaw splitting strength of PU-modified asphalt at different ages was obtained, and it was found that the PU content was the most significant.(3)The optimal ratio parameters of PU-modified asphalt are determined as follows: the PU content is 56.64%, the preparation process is the prefabricated method, and the CA content is 3.58%. The establishment of the response surface was found to be highly consistent, which can therefore guide the optimization of the PU-modified asphalt ratio.(4)Through conducting tensile tests, it was found that the PU asphalt prepared by the best scheme has good tensile properties and flexibility.(5)Through the analysis of failure surface morphology, it was found that PU-modified asphalt significantly improves the strength of the binder–aggregate interface, and the material performance was found to be excellent. PU-modified asphalt exhibits obvious strengthening and toughening effects on the mixture.

## Figures and Tables

**Figure 1 polymers-15-02327-f001:**
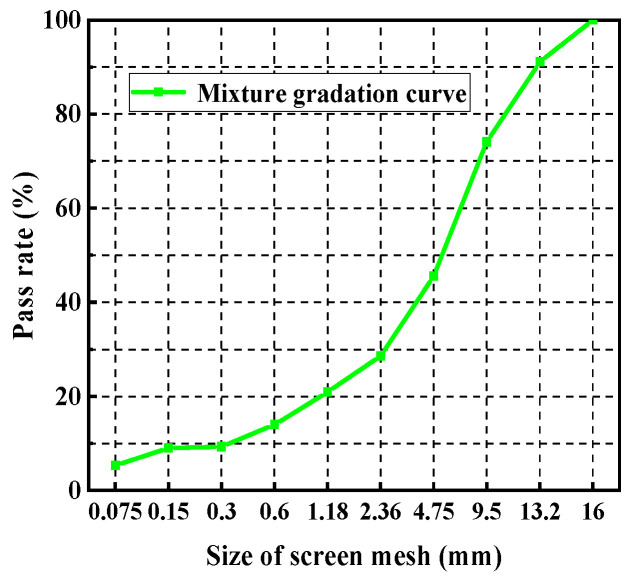
Mixture gradation curve.

**Figure 2 polymers-15-02327-f002:**
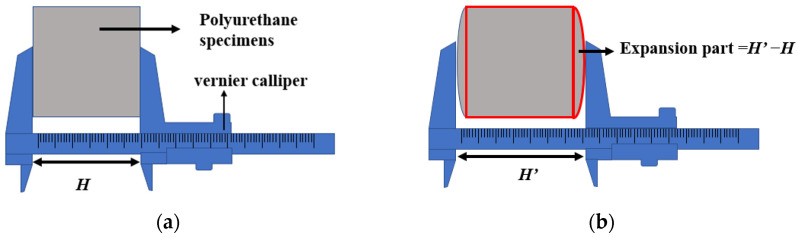
(**a**) Before specimen expansion; (**b**) After specimen expansion.

**Figure 3 polymers-15-02327-f003:**
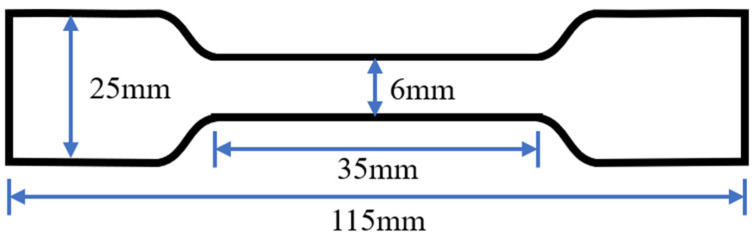
Marshall specimen forming process.

**Figure 4 polymers-15-02327-f004:**
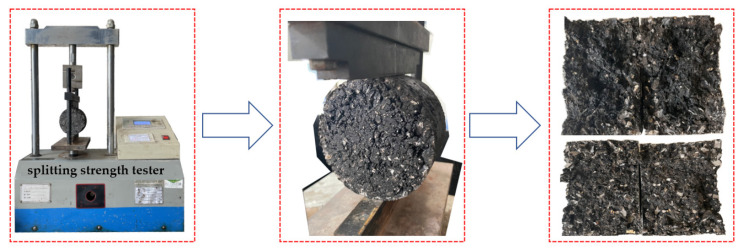
Marshall specimen splitting strength test.

**Figure 5 polymers-15-02327-f005:**
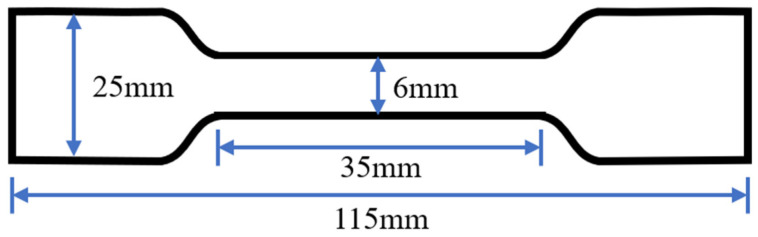
Tensile test specimen size.

**Figure 6 polymers-15-02327-f006:**
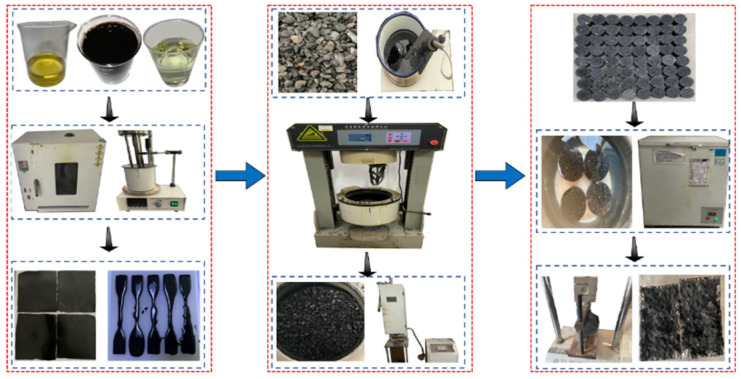
Test process flow chart.

**Figure 7 polymers-15-02327-f007:**
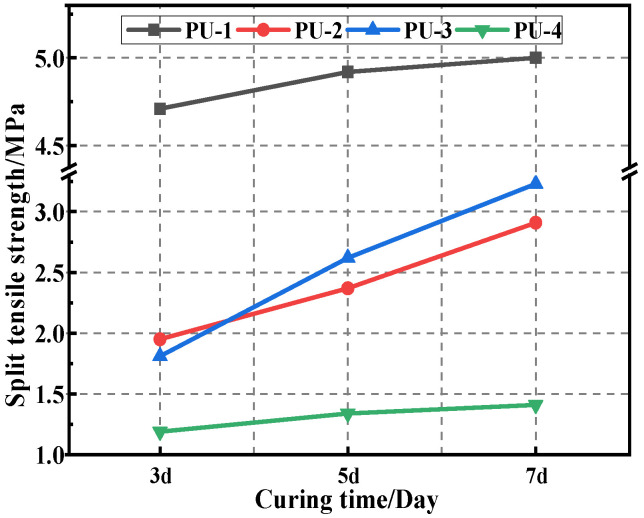
Splitting strength results.

**Figure 8 polymers-15-02327-f008:**
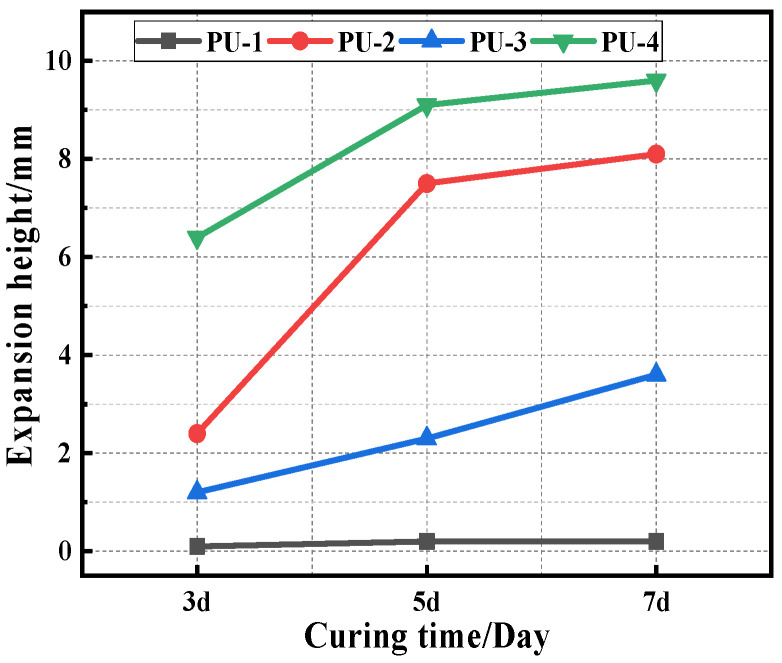
Expansion heights of the specimens.

**Figure 9 polymers-15-02327-f009:**
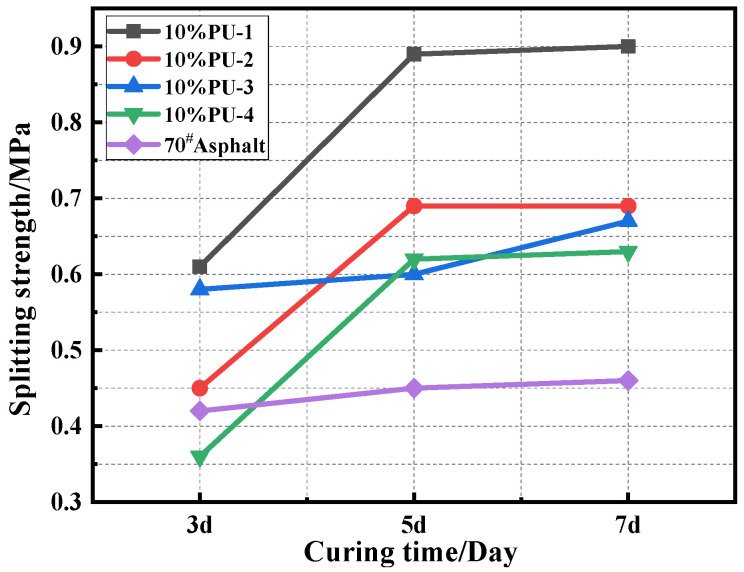
Splitting strengths of the different PU asphalt mixtures.

**Figure 10 polymers-15-02327-f010:**
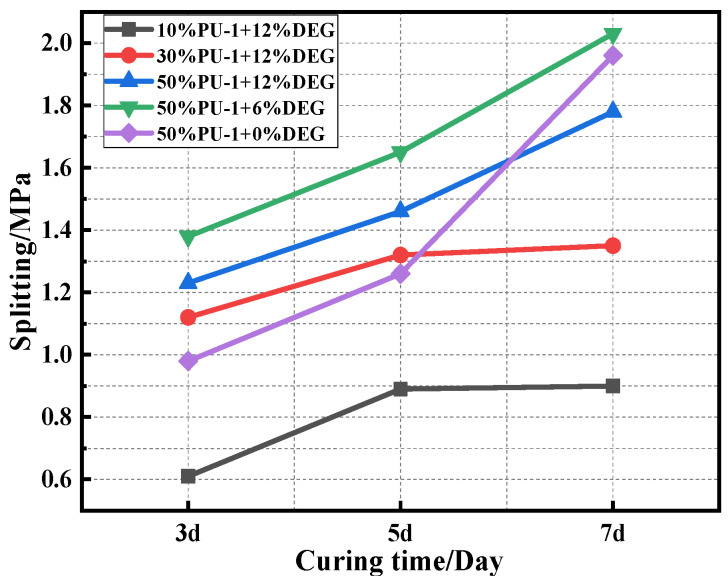
Splitting strength of the PU asphalt mixtures with various contents.

**Figure 11 polymers-15-02327-f011:**
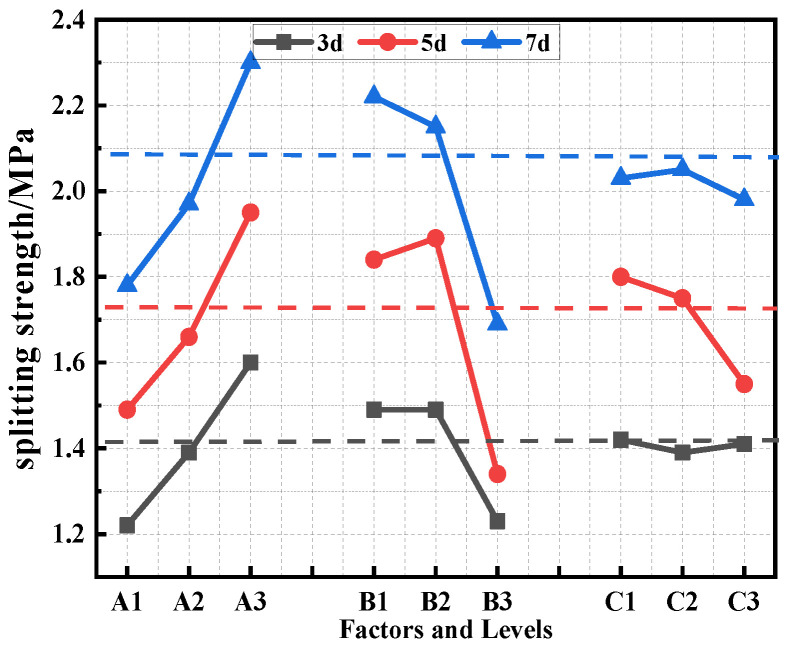
Effects of the various factors on the splitting strength of the specimens.

**Figure 12 polymers-15-02327-f012:**
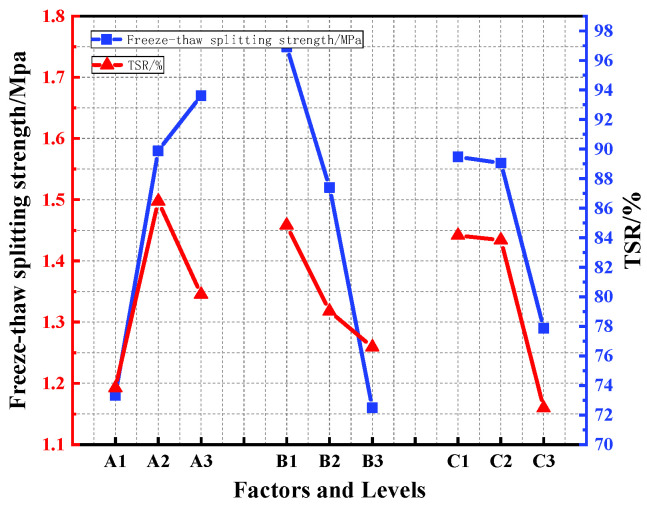
Effects of the various factors on the freeze-thaw splitting strength and TSR of the specimens.

**Figure 13 polymers-15-02327-f013:**
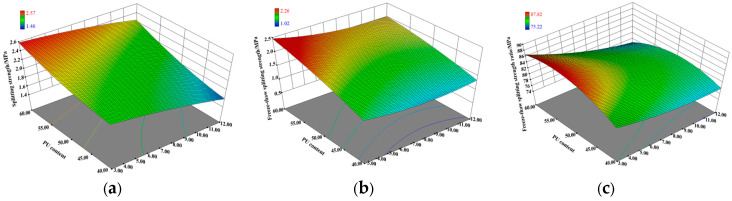
Response surface: (**a**) splitting strength; (**b**) freeze-thaw splitting strength; and (**c**) TSR.

**Figure 14 polymers-15-02327-f014:**
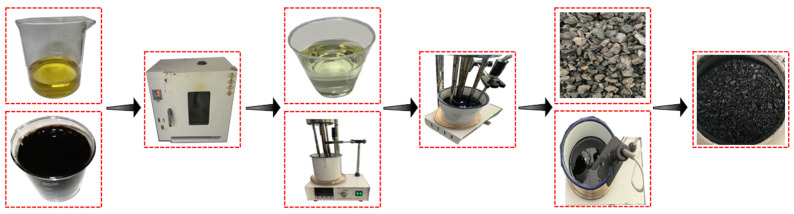
Optimal scheme flow chart.

**Figure 15 polymers-15-02327-f015:**
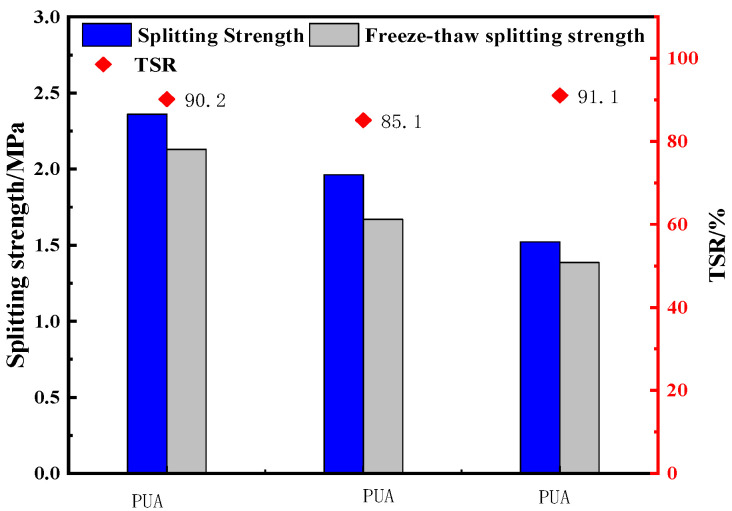
Performance comparison of the mixture.

**Figure 16 polymers-15-02327-f016:**
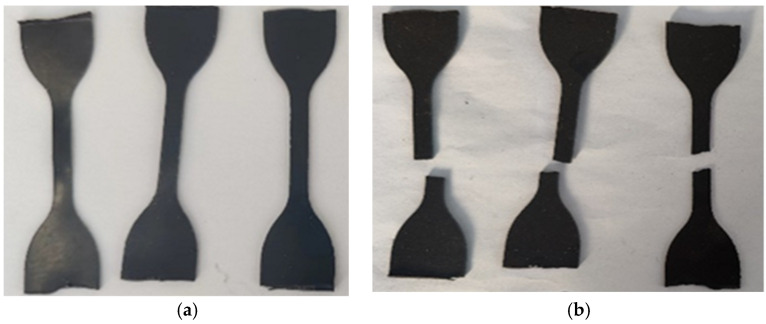
Tensile test specimen: (**a**) shape before tensile test; and (**b**) shape after tensile test.

**Figure 17 polymers-15-02327-f017:**
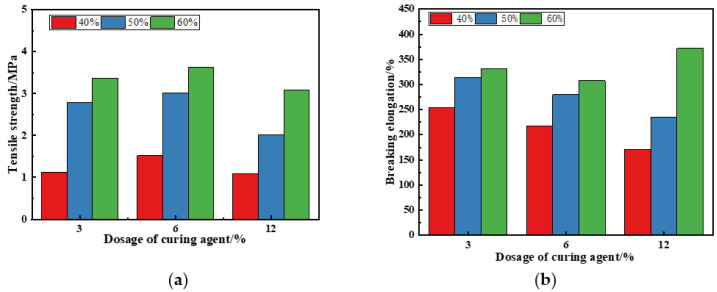
Tensile properties of PU asphalt: (**a**) tensile strength; and (**b**) breaking elongation.

**Figure 18 polymers-15-02327-f018:**
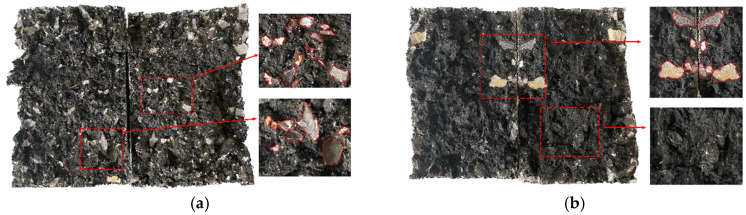
(**a**) Splitting surface diagram of the PU asphalt mixture; and (**b**) freeze-thaw splitting surface of the PU asphalt mixture.

**Figure 19 polymers-15-02327-f019:**
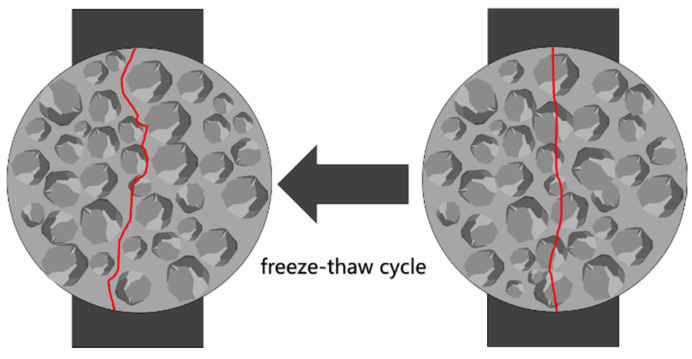
Damage simulation diagram of the PU asphalt mixture specimen.

**Table 1 polymers-15-02327-t001:** Base asphalt technical indicators.

Performance Indicators	Penetration (25 °C)/0.1 mm	Ductility (10 °C)/cm	Softening Point/°C	Flash Point/°C	Wax Content/%	Density (15 °C)/(g·cm^−3^)
Test results	65.3	56	48	330	203	1.036

**Table 2 polymers-15-02327-t002:** Aggregate density summary.

Aggregate Size	Apparent Relative Gravity/γa	Relative Density of Bulk Volume/γb	Water Absorption (%)	Asphalt Absorption Coefficient	Effective Relative Density (g/cm^3^)
Limestone 12–16 mm	3.315	3.227	1.10	0.38	3.307
Limestone 6–12 mm	2.935	2.823	1.33	0.59	2.895
Limestone 4–6 mm	2.773	2.721	1.68	0.79	2.715
Limestone 0–4 mm	2.692	2.573	1.80	0.92	2.687
Mineral powder	2.636	2.636	-	-	2.636

**Table 3 polymers-15-02327-t003:** PU modifier raw material description.

Numbering	Type	CA Type	Tensile Strength/MPa	Breaking Elongation/%
GXJ-1	PU prepolymer	Small molecule polyol (CA-1)	11.9	330
GXJ-2	PU prepolymer	Small molecule polyol (CA-1)	20.4	183
GXJ-3	Castor oil	Isocyanate (CA-2)	15.2	80
GXJ-4	Polyether polyol	Isocyanate (CA-2)	18.6	245

**Table 4 polymers-15-02327-t004:** AC-13 asphalt mixture component material.

Numbering	Gradation Type	PU Type	OAC/%	VV/%	VMA/%	VFA/%
PU asphalt	AC-13	GXJ-1	6.1	4.1	14.9	72.5
70#A asphalt	AC-13	-	4.5	4.3	15.3	71.89

**Table 5 polymers-15-02327-t005:** Orthogonal factors and horizontal.

Orthogonal Factors	PU Content (A)	Preparation Method (B)	CA Content (C)
1	40%	Prefabrication method	3%
2	50%	Post-doping method	6%
3	60%	Pre-reaction method	12%

**Table 6 polymers-15-02327-t006:** Orthogonal design table.

Numbering	Orthogonal Composite	A	B	C
1	A1B1C3	40	1	12
2	A1B2C1	40	2	3
3	A1B3C2	40	3	6
4	A2B1C1	50	1	3
5	A2B2C2	50	2	6
6	A2B3C3	50	3	12
7	A3B1C2	60	1	6
8	A3B2C3	60	2	12
9	A3B3C1	60	3	3

**Table 7 polymers-15-02327-t007:** Introduction to the PU asphalt preparation process.

Preparation Method	Description
Prefabrication method	The base asphalt is sheared with polyurethane. After the shearing process, the CA is added and then mixed with the aggregate.
Post-doping method	The base asphalt is sheared with the CA; after the shearing process, it is mixed with the aggregate, and polyurethane is added during the mixing process.
Pre-reaction method	The polyurethane is sheared with the CA. Following the shearing process, the base asphalt is added and then mixed with the aggregate.

**Table 8 polymers-15-02327-t008:** Results of the orthogonal tests.

Number	Orthogonal Composite	3d Split Strength/MPa	5d Split Strength/Mpa	7d Split Strength/Mpa	Freeze-Thaw Splitting Strength/Mpa	TSR/%
1	A1B1C3	1.32	1.44	1.93	1.12	64.48
2	A1B2C1	1.23	1.74	1.96	1.4	79.37
3	A1B3C2	1.12	1.28	1.46	1.02	77.63
4	A2B1C1	1.57	1.97	2.26	1.98	87.82
5	A2B2C2	1.47	1.86	2.11	1.57	82.68
6	A2B3C3	1.13	1.14	1.64	1.15	77.91
7	A3B1C2	1.58	2.10	2.57	2.01	83.52
8	A3B2C3	1.77	2.08	2.37	1.8	75.01
9	A3B3C1	1.46	1.68	1.96	1.31	74.26

**Table 9 polymers-15-02327-t009:** Variance analysis of the orthogonal experiment.

Response Value		PU Content	Preparation Method	Dosage of CA	Error Value
3d Splitting strength	SS	0.218	0.128	0.001	0.046
	d/f	2	2	2	2
	MS	0.109	0.064	0.001	0.023
	F	4.77	2.81	0.03	
5d Splitting strength	SS	0.335	0.501	0.099	0.027
	d/f	2	2	2	2
	MS	0.167	0.251	0.050	0.014
	F	12.29	18.42	3.64	
7d Splitting strength	SS	0.411	0.501	0.007	0.007
	d/f	2	2	2	2
	MS	0.205	0.251	0.003	0.003
	F	60.20	73.50	1.03	
Freeze-thaw splitting strength	SS	0.410	0.526	0.157	0.093
	d/f	2	2	2	2
	MS	0.205	0.263	0.078	0.046
	F	4.43	5.68	1.69	
TSR	SS	0.024	0.011	0.027	0.018
	d/f	2	2	2	2
	MS	0.012	0.005	0.013	0.009
	F	1.31	0.59	1.46	

**Table 10 polymers-15-02327-t010:** Variance and reliability analysis of the three response values.

Response Value		Quadratic Sum	Degree of Freedom	Mean Square	F Ratio	*p*-Value (>F) ^1^
7d Splitting Strength	Model	0.926	6	0.003	14.88	0.0002
	PU content	0.411	2	0.205	22.83	0.0003
	Preparation method	0.501	2	0.251	15.42	0.0017
	Dosage of CA	0.007	2	0.004	6.38	0.0253
Freeze-thaw splitting strength	Model	1.87	6	0.21	19.67	0.0001
	PU content	0.60	2	0.60	56.86	0.0002
	Preparation method	0.78	2	0.78	73.79	0.0001
	Dosage of CA	0.23	2	0.23	21.62	0.0109
TSR	Model	199.36	6	22.15	35.52	0.0001
	PU content	8.41	2	8.41	13.49	0.0079
	Preparation method	11.98	2	11.98	19.21	0.0032
	Dosage of CA	111.30	2	111.30	178.48	0.0001

^1^ When the *p*-value is greater than 0.05, the model significance is low (indicating no statistical significance), and the regression model is not available; if the converse is true, the regression model significance is high.

**Table 11 polymers-15-02327-t011:** Optimal composition.

PU Content	Preparation Method	Dosage of CA
56.64%	Prefabrication method	3.58%

## Data Availability

This original copy does not include distributed figures and tables, thus all figures and tables in this original copy are unique.

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
