# Peer review of "Preparation Scheme Optimization of Thermosetting Polyurethane Modified Asphalt"

_polymers, 2023, doi:10.3390/polym15102327_

Round 1
Reviewer 1 Report
The article concerns an interesting and necessary direction of development of road materials, which is the need to replace traditional bituminous binders with modern synthetic binders.
The authors presented extensive research results and proposed the optimal composition of the binder.
It should be remembered that laboratory results are presented, and in a case as important as the development of a new binder, tests at the asphalt plant in real conditions and field tests are necessary.
Please refer to some minor remarks:
1. The mineral mix consists of coarse limestone aggregate - is this solution common in China? In my our region, coarse limestone aggregate is not allowed for wearing course.
2. The term 'matrix asphalt' is misleading. If I understand correctly, it's about asphalt binder. 'Stone matrix asphalt' is a kind of mineral-asphalt mixture (stone mastic asphalt).
3. line 159 -it is better to use 'test specimen' instead of 'test piece'
4. Numbering and descriptions of figures should be corrected.
5. More information about the asphalt mix should be provided: what type of mix (AC - asphalt concrete?), asphalt binder content, compaction details (Marshall hammers - how many blows, compation temperature, short term ageing before compaction?), content of voids in Marshall samples.
6. Line 213 - is the tensile rate of 500 mm/min correct? in typical asphalt binder testing, the tensile rate is 50 mm/min
7. There is no comparison to mixtures with road bitumen, or SBS modified bitumen. Do the authors have test results for an analogous mixture with traditional binders?
Reviewer 2 Report
In this study, the authors present the optimization of thermosetting polyurethane modified asphalt. Generally speaking, this is a very interesting paper. Many experiments were conducted and the conclusions are well supported by the results. Below are my comments.
1. The first paragraph of introduction should also include the benefits of modified asphalt including the enhanced moisture damage resistance. (e.g., "Moisture damage mechanism and thermodynamic properties of hot-mix asphalt under aging conditions. ACS Sustainable Chemistry & Engineering, 10(45), pp.14865-14887."). The recently proposed graphene modification should be included.
2. I suggest that the first sentence of each paragraph should summarize the main idea of the whole paragraph. If not, readers are easy to get lost.
3. Do you have any information about asphalt mix design? The volumetric properties of asphalt mixtures are not known.
4. I suggest the authors use TSR (tensile strength ratio) instead of freeze-thaw splitting tensile strength ratio.
5. The section 3.6 should be rewritten. More indepth discussion should be made according to the test results.
Some English expressions are understandable but kind of weird and not natural. Please double check and further improve the quality of English.
Round 2
Reviewer 2 Report
This paper has been revised based on the comments and should be ready for publication.